# Impact of MASLD on Portal Vein Thrombosis Following Hepatectomy for Liver Cancer

**DOI:** 10.3390/cancers16223844

**Published:** 2024-11-15

**Authors:** Yoshito Wada, Koji Okuda, Shin Sasaki, Shigeo Shimose, Takamichi Nishida, Hisaaki Shimokobe, Yuichi Nagao, Takayuki Torigoe, Koji Hayashi, Hidetoshi Akashi, Satoshi Taniwaki, Tetsuo Imamura

**Affiliations:** 1Department of Surgery, Kyoaikai Tobata Kyoritsu Hospital, Kitakyushu 804-0093, Fukuoka, Japan; yo-wada@kyoaikai.com (Y.W.); sasaki_shin@med.kurume-u.ac.jp (S.S.); nishida_takamichi@kurume-u.ac.jp (T.N.); md230009@cis.fukuoka-u.ac.jp (H.S.); y-nagao@med.uoeh-u.ac.jp (Y.N.); torigoe@med.uoeh-u.ac.jp (T.T.); ko-hayashi@kyoaikai.com (K.H.); hi-akashi@kyoaikai.com (H.A.); taniwaki@kyoaikai.com (S.T.); imamura@kyoaikai.com (T.I.); 2Department of Internal Medicine, Tobata Kyoritsu Hospital, Kitakyushu 804-0093, Fukuoka, Japan; shimose_shigeo@med.kurume-u.ac.jp

**Keywords:** non-alcoholic fatty liver disease, venous thrombotic risk, metabolic dysfunction-associated steatotic liver disease, portal vein thrombosis, hepatectomy, hepatic steatosis

## Abstract

The impact of metabolic dysfunction-associated steatotic liver disease (MASLD) on post-hepatectomy portal vein thrombosis (PH-PVT) was investigated. One hundred six patients who underwent hepatectomy were categorized based on the presence of liver steatosis as MASLD (*n* = 20), MetALD (MASLD with moderate alcohol consumption) (*n* = 5), Other SLD (hepatic steatosis of other etiology) (*n* = 13), and No SLD (no hepatic steatosis) (*n* = 68). PH-PVT was observed in twelve patients. The incidences were 35.0% in MASLD, 20.0% in MetALD, 7.7% in Other SLD, and 4.4% in No SLD. The MASLD group (including MASLD and MetALD) was an independent significant risk factor for PH-PVT. Moreover, the incidence of MASLD was significantly higher than that of No SLD, along with metabolic factors, excluding alcohol consumption. In conclusion, MASLD and MetALD are risk factors for PH-PVT, and hepatic steatosis and metabolic dysfunction may play a synergistic role in the development of PH-PVT.

## 1. Introduction

Hepatectomy is a common treatment for primary and metastatic liver tumors originating from colorectal and neuroendocrine tumors. Recent advancements in hepatectomy techniques and perioperative care have reduced postoperative complications and mortality rates. However, complications still occur in a significant number of patients, ranging from 18% to 48% [1,2]. Post-hepatectomy liver failure, bile leakage, intraperitoneal abscesses, pulmonary complications, and venous thrombosis are major complications that can lead to mortality. Post-hepatectomy portal vein thrombosis (PH-PVT) is a potentially life-threatening complication, which is usually asymptomatic but can lead to severe liver damage due to reduced portal flow and portal hypertension as it progresses.

The global prevalence of steatotic liver disease (SLD), particularly non-alcoholic fatty liver disease (NAFLD), has recently increased [3]. Therefore, there has been a rise in the number of patients undergoing hepatectomy for underlying NAFLD. Although NAFLD is closely linked to metabolic disorders, in 2023, a multi-societal effort addressed the limitations of the definition of NAFLD and updated it to metabolic dysfunction-associated steatotic liver disease (MASLD) to accurately reflect the underlying pathophysiology [4,5]. MASLD requires a new inclusion criterion of “the presence of at least one or more cardiometabolic risk factors.”

Studies must build on the definition of MASLD to maximize the utilization of research resources, as this revision has been supported by several reports [6]. Metabolic disorders such as obesity and type 2 diabetes mellitus, as well as NAFLD itself, lead to an inflammatory milieu and prothrombotic conditions, and these environments increase venous thrombotic risk [7,8]. However, there is no evidence of the effects of metabolic disorders and MASLD/NAFLD on PH-PVT. We, therefore, aimed to investigate the effect of MASLD on PVT after hepatectomy for liver cancer.

## 2. Patients and Methods

### 2.1. Study Design and Patients

This study was approved by the Institutional Review Board of the Kyoaikai Tobata Kyoritsu Hospital (approval number: 24-04). The retrospective study included 129 consecutive patients who underwent hepatectomy between June 2011 and December 2023 at Kyoaikai Tobata Kyoritsu Hospital. The exclusion criteria were as follows: (1) age < 16 years (*n* = 0); (2) hepatectomy for benign disease (*n* = 6); (3) concomitant resection of the extrahepatic bile duct or gastrointestinal tract (*n* = 13); (4) resection and reconstruction of the portal vein or hepatic artery (*n* = 0); and (5) lack of enhanced CT imaging during the perioperative period to identify PH-PVT (*n* = 4) (Figure 1). We excluded patients with benign tumors. Malignant tumors are prone to hypercoagulable conditions [9]; thus, benign and malignant tumors may have different risk factors for venous thrombosis. We also excluded patients who were unable to undergo enhanced CT examinations; all of these patients had renal insufficiency or allergic reactions to the iodine contrast medium. This exclusion was necessary because diagnosing the presence or absence of PH-PVT in the early postoperative phase is challenging without the enhanced CT findings.

### 2.2. Diagnosis and Definition of SLD Categories

On the preoperative non-contrast CT scan image, the mean Hounsfield unit (HU) value was measured in the maximum circular regions of interest, which were at least 1 cm^2^. These measurements were obtained from two points on the right liver lobe (anterior and posterior sectors), one point on the left liver lobe, and one point on the spleen. The ratio of the average HU values of the liver to that of the spleen (L/S ratio) was calculated. An L/S ratio of less than 1.1 indicates a positive diagnosis of SLD [10,11].

In the presence of SLD, the categories complied with the criteria recently defined by the Delphi consensus [4,5]. MASLD was defined as SLD in conjunction with at least one of five cardiometabolic factors: (1) obesity: BMI ≥ 23 kg/m^2^ or waist circumference ≥ 85 cm for men, ≥90 cm for women (Japanese); (2) hyperglycemia: fasting serum glucose ≥ 100 mg/dL, or 2-h post-load glucose levels ≥ 140 mg/dL, or HbA1c ≥ 5.7%, or type 2 diabetes, or treatment for type 2 diabetes; (3) hypertension: blood pressure ≥ 130/85 mmHg, or specific antihypertensive drug treatment; (4) increased plasma triglycerides: plasma triglycerides 150 mg/dL, or lipid-lowering treatment; and (5) reduced plasma high-density lipoprotein-cholesterol, plasma: HDL-cholesterol ≤ 40 mg/dL for men, ≤50 mg/dL for females, or lipid-lowering treatment. Further, MASLD without alcohol consumption was referred to as “MASLD”, and MASLD with moderate alcohol consumption (30–60 g/day or 210–420 g/week for men, 20–50 g/day or 140–350 g/week for women) was referred to as “MetALD”. This threshold of alcohol consumption distinguishing MASLD from Met ALD was outlined by Rinella et al. in the multi-society Delphi consensus statement on the new nomenclature for fatty liver disease [5] Other entities under SLD included alcohol-related liver disease (ALD) (regular drinking >60 g/day for men and 50 g/day for women), specific etiologies of SLD (drug-induced steatosis, viral hepatitis, autoimmune hepatitis, monogenic diseases, miscellaneous), and cryptogenic SLD, collectively referred to as “Other SLD”. An L/S ratio < 1.1 was represented as “No SLD”.

### 2.3. Surgical Procedure

Hepatic parenchymal transection was performed using an ultrasonically activated scalpel (SonoSurg, Olympus, Tokyo, JAPAN) or the clamp–crush technique in cases of open laparotomy and only the clamp–crush technique in cases of the laparoscopic approach. Before the transection, we performed extrahepatic Glissonean pedicle ligation for mono and central sectionectomies. Individual ligation of the hepatic vessels was used in hemihepatectomy, extended hemihepatectomy, and trisectionectomies [12]. During parenchymal transection, we routinely applied intermittent inflow occlusion with 15 min of occlusion, followed by five min of reperfusion, repeated as needed, using Pringle’s maneuver with the Rummel tourniquet technique. We classified the type of resection into “partial resection”, “left lateral sectionectomy”, and “anatomical resection”. We defined atypical resections and wedge resections as “partial resection”, while “anatomical resection” refers to systematic resections of Couinaud segments, sections, the right and left hemiliver, and the right and left trisection, following the Brisbane 2000 system of nomenclature for hepatic anatomy and resections [12]. However, we excluded left lateral sectionectomy from the “anatomical resection” category, placing it in an independent category instead. Before hepatectomy, if the future remnant liver volume was insufficient, portal vein embolization (PVE) was performed on liver segments affected by the resected liver. This procedure involved using absolute ethanol as an embolization material, administered via the transileocolic route, and was conducted 2–4 weeks prior to hepatectomy [13,14].

### 2.4. Identifying and Managing PH-PVT

Identification of PH-PVT was assessed using contrast-enhanced CT images at four different points in the course: within two months before surgery, four to seven days after surgery, and approximately three and six months after surgery. When PH-PVT was identified, anticoagulation therapy was promptly initiated with systemic heparinization with or without AT III supplementation, followed by oral warfarin or direct oral anticoagulants (DOAC). Few patients were treated with warfarin or DOAC alone.

### 2.5. Data Collection

The following data were collected from the medical records: age, sex, alcohol history, diagnosis of liver tumor, underlying liver disease, comorbidities, preoperative use of oral anticoagulants and other medicines for metabolic disorders, serum levels of alanine transaminase (ALT), total bilirubin, albumin, platelet count, prothrombin time (%), Child-Pugh score (points), maximum size and number of tumors, presence of portal vein tumor thrombus, presence of preoperative PVE, type of hepatectomy, hepatectomy with open or laparoscopic approach, operative time, intraoperative blood loss volume, and times of the intermittent vascular occlusion. The types of hepatectomies are classified as partial resection, left lateral sectionectomy, and anatomical resection, including one or more liver segments. Postoperative complications were classified according to the Clavien–Dindo classification [15].

### 2.6. Statistical Analyses

Data were expressed as mean ± standard deviations and ranges, while categorical variables were presented as frequencies and percentages. All statistical analyses were performed using JMP Pro version 16 (SAS Institute Inc., Cary, NC, USA). Differences in the variables between the presence and absence of PH-PVT were analyzed using Student’s t-test, Pearson’s correlation test, and logistic regression analysis.

Explanatory variables were selected stepwise to investigate independent factors associated with PH-PVT, minimizing the Bayesian information criterion as previously described [16,17]. The following factors were applied in the stepwise procedure: age; sex; disease; SLD category; total bilirubin, ALT, and albumin levels; prothrombin time; platelet count; Child-Pugh score (points); tumor size; tumor number; macroscopic vascular invasion; previous chemotherapy; previous anticoagulant; preoperative PVE; type of hepatectomy; operative approach; operation time; intraoperative bleeding; Pringle maneuver; blood transfusion; concomitant RFA; and postoperative complications excluding PH-PVT. The selected variables were used in a logistic regression analysis to identify independent factors for PH-PVT.

The level of statistical significance was set at *p* < 0.05. We also performed a decision tree analysis to identify factors associated with PH-PVT, as previously described [18]. Furthermore, to investigate the impact of the sole metabolic disorder on PH-PVT, the difference in the incidence of PH-PVT was analyzed between patients with MASLD and patients without SLD, along with metabolic factors, excluding alcohol consumption.

## 3. Results

### 3.1. Patient Characteristics

Twenty-three patients were excluded from this study, and 106 were enrolled (Figure 1). The clinical characteristics of the patients are summarized in Table 1. Thirty-eight patients had SLD (38.7%). MASLD was defined in 20 patients, MetALD in five, Other SLD in 13 (ALD in seven, viral hepatitis in five, cryptogenic SLD in one), and No SLD in 68 patients. Sixty patients had primary liver cancer, and 46 had metastatic liver cancer. Twenty-four patients received systemic chemotherapy or locoregional chemotherapy via the hepatic artery within six months of surgery. These patients underwent resection more than one month after the final chemotherapy treatment. Twenty patients previously underwent oral anticoagulant therapy owing to comorbidities such as arrhythmia, cardiac disease, and cerebrovascular disease. These oral anticoagulants were discontinued before surgery in a usual manner to avoid increased intraoperative bleeding. For these patients, anticoagulant therapy was resumed three to five days postoperatively. The other patients did not receive any postoperative prophylactic anticoagulant therapy to prevent venous thrombosis. PVE was performed in 10 patients prior to the surgery.

### 3.2. Perioperative Courses

The operative and postoperative outcomes are presented in Table 2. Partial resection was performed in 40 patients, 15 underwent left lateral sectionectomy, and 51 underwent anatomical resection. Laparoscopic surgery was performed in 29 patients, and concomitant radiofrequency ablation was performed in nine patients. PH-PVT was found in 12 out of 106 patients (11.3%). One patient with PH-PVT had concomitant hepatic vein and inferior vena cava thromboses. Regarding other thrombotic complications, one patient had deep vein thrombosis, and the other had pulmonary embolism without any symptoms. Among patients with PH-PVT, the thrombus was identified in the first branch of the left or right portal vein in three patients, in the umbilical portal vein in eight patients, and in the third branch of the left lobe in one patient. None of the patients had thrombus in the main portal vein or beyond. The stenosis rates were categorized as follows: less than 50% in two patients, 50% to 70% in four patients, 70% to 90% in three patients, and over 90% in three patients (Table 3).

### 3.3. Analysis of Risk Factors of PH-PVT

In the univariate analysis, SLD category and type of hepatectomy were statistically significant risk factors for PH-PVT (Table 4). By stepwise regression, SLD category ([MASLD and MetALD] vs. [Other SLD and No SLD]) and type of hepatectomy ([left lateral sectionectomy] vs. [partial and anatomical resection]) were selected and revealed that MASLD and MetALD (odds ratio 9.27, 95% CI 2.32–37.08, *p* = 0.0016) and left lateral sectionectomy (odds ratio 6.22, 95% CI 1.40–27.67, *p* = 0.0165) were independent risk factors for PH-PVT (Table 5), determined using a logistic regression analysis. A decision tree analysis was conducted to identify the clinical profile associated with PH-PVT. The SLD category was determined to be the primary distinguishing factor for PH-PVT. PH-PVT was observed in 32% of patients with MASLD and MetALD. In contrast, PH-PVT was observed in 4.9% of patients with Other SLD and No SLD. Among patients with MASLD and MetALD, the type of hepatectomy was the second most significant distinguishing factor. PH-PVT was observed in 60% of patients who underwent left lateral sectionectomy compared with 25% of patients who underwent partial or anatomical resection (Figure 2). The incidence of PH-PVT was compared between patients with MASLD and those without SLD associated with cardiometabolic factors (excluding alcohol consumption) to investigate the impact of the sole metabolic disorder on PH-PVT. PH-PVT incidence in No SLD with cardiometabolic factors, excluding alcohol consumption, was 2.6%, significantly lower than the 35.0% observed in MASLD (*p* = 0.0006) (Table 6).

### 3.4. Treatments and Outcomes of PH-PVT

Ten of the twelve patients diagnosed with PH-PVT received anticoagulant therapy immediately after identification. This included systemic heparinization with or without AT III supplementation, followed by warfarin or direct oral anticoagulants in seven patients. One patient received only low-molecular-weight heparin, another only warfarin, and one only a direct oral anticoagulant. Two patients did not receive anticoagulant therapy: one had an anaphylactic reaction to anticoagulants; the other developed PH-PVT that went unnoticed during the postoperative period. PH-PVT was later identified through a retrospective examination of CT images for this study. In all patients, including those who received no anticoagulant therapy, PH-PVT completely disappeared within six months after hepatectomy without invasive thrombectomy.

## 4. Discussion

The prevalence of PH-PVT in this study was 11.3%, consistent with previous studies reporting a range of 4% to 23% [19,20]. However, the risk factors for PH-PVT have not yet been thoroughly elucidated. Patients undergoing major right hepatectomy, left lateral sectionectomy, longer Pringle maneuver time, longer operation time, liver cirrhosis, or intraoperative blood transfusion have a high incidence of PH-PVT [19,20,21]. We found that MASLD and MetALD were independent risk factors for PVT and left lateral sectionectomy.

The pathophysiology of venous thrombosis encompasses one or more features of the Virchow triad: reduced venous blood flow, a hypercoagulable state, and vascular endothelial injury [22]. NAFLD increases the risk of venous thromboembolism, which is 2.5 times higher compared to other liver diseases [23]. However, the mechanism underlying increased thrombotic risk in patients with NAFLD remains unclear. Several basic, translational, and clinical studies have postulated that repetitive injury from chronic inflammation due to hepatic steatosis and lipid deposition over time leads to endothelial cell activation, oxidative injury, and necroapoptosis. Additionally, these inflammatory processes may lead to an imbalance in hemostasis, thereby disrupting the delicate balance of hemostasis and favoring hypercoagulability such as platelet activation, an increase in factor VIII, an increase in plasminogen activator inhibitor-1, and a decrease in protein C [24,25]. However, a recent report by Potze et al. challenged this paradigm, suggesting that NAFLD has a limiting role for hyperactive hemostasis in increasing thrombotic risk and that obesity, which is closely linked to SLD, was more strongly associated with a hypercoagulable environment than NAFLD itself [26]. Despite these extensive pathophysiological studies, no study has investigated the impact of SLD and metabolic dysfunction on PH-PVT development.

Recently, a consensus was reached regarding hepatic steatosis and metabolic dysfunction. NAFLD should be updated to MASLD to reflect the underlying pathophysiology accurately [5]. The process of this revision was led by the American Association for the Study of Liver Disease (AASLD), the European Association for the Study of the Liver (EASL), and the Association Latinoamericana para el Estudio del Higado (ALEH). Several studies have presented evidence supporting this transition. Therefore, it is essential to focus on MASLD rather than NAFLD [6,27]. In the definition of MASLD, hepatic steatosis with cardiometabolic factor was categorized into MASLD and MetALD. MASLD is characterized by the absence of alcohol consumption, whereas MetALD involves moderate alcohol consumption. Despite reports of the differing clinical significance between the two groups, specific results have not yet been obtained [28,29]. In the current study, due to the small number of MetALD cases, we were unable to assess the difference between the two groups confidently. As a result, MASLD and MetALD were combined into one group and analyzed using multivariate analysis in this study.

The incidence of PH-PVT in patients with MASLD and MetALD was significantly higher than in patients with other causes of SLD (Other SLD) and those without SLD (No SLD). Furthermore, to investigate the impact of the sole metabolic disorder, we compared the incidence of PH-PVT between MASLD and No SLD associated with metabolic dysfunction with alcohol consumption. The results showed a significantly lower incidence in patients with No SLD associated with metabolic dysfunction with alcohol consumption. These results indicate that the combination of steatosis and metabolic dysfunction creates a significant procoagulant condition that contributes to the development of PH-PVT and cannot be attributed to metabolic dysfunction or hepatic steatosis alone. To the best of our knowledge, this is the first report of this phenomenon.

PH-PVT is also preferred in the umbilical portion of the portal vein, particularly after a left lateral sectionectomy. It has been speculated that the blood flow in the umbilical portion might be comparatively lower than the vessel size following left lateral sectionectomy. In addition, blood flow in the umbilical portion slows down physiologically when in the supine position [21]. Furthermore, the round ligament of the liver was pulled to flatten the cutting plane during the surgical procedure of left lateral sectionectomy. This procedure may induce strong tension in the umbilical portion, causing vascular endothelial injury [21]. However, the definitive reason for this has not yet been clarified.

Few studies have reported the natural course of PH-PVT. In cases with cirrhosis, Luca et al. reported that portal vein thrombosis was worsened in 48% of the cases and improved in 45% without anticoagulation therapy [30]. Regarding the validity of anticoagulant therapy, Condat et al., in their study of acute portal vein or mesenteric vein thrombosis in various diseases, described that recanalization occurred in 25 of 27 patients administered anticoagulation and in 0 of 2 patients not administered anticoagulation [31]. We promptly initiated anticoagulant therapy to prevent thrombosis extension and reduce the risk of liver failure due to decreased portal venous flow or portal hypertension. However, anticoagulant therapy was not administered in two patients. One patient experienced an anaphylactic response to anticoagulants. In another patient, the development of PH-PVT was not pointed out during the postoperative course, whereas PH-PVT was identified by retrospective inspection of CT images. In all patients, including those who did not receive anticoagulant therapy, PH-PVT completely disappeared within six months after hepatectomy, and none of the patients required invasive treatments, such as operative thrombectomy. The effectiveness of early treatment is likely to be higher than that of late treatment after the development of PVT because the thrombus has organized and adhered to the vascular wall later [32]. There was no difference in the anticoagulant effectiveness between patients with and without MASLD.

This study had several limitations. First, this was a retrospective, single-center study with a small sample size. In this study, previously reported risk factors for PH-PVT, such as operative time, operative bleeding volume, duration of Pringle maneuvers, and major hepatectomy, including right hemi-hepatectomy and tri-sectionectomy, were not found to be risk factors. Moreover, we could not assess the difference between MASLD and MetALD, which refers to the impact of moderate alcohol consumption on PH-PVT, primarily because of the limited number of patients with MetALD. These issues should be studied with larger sample sizes. Second, the diagnosis of steatotic liver was predicted based on the L/S ratio of CT imaging. This diagnostic method for steatotic livers has provided clear evidence [10,11], and we complied with this diagnostic approach for SLD. The definition of MASLD proposed by the AASLD does not require obligatory liver biopsies [4]. However, future studies on histologically proven SLD would be helpful in evaluating the relationship between the pathophysiological or grade of steatosis/steatohepatitis and the development of PH-PVT. Third, several patterns of anticoagulant treatment were administered for PH-PVT in this study, as anticoagulant treatment guidelines have not yet been established. Further studies are needed to develop indications and standard anticoagulant therapy to verify the efficacy of anticoagulant therapy for PH-PVT.

## 5. Conclusions

MASLD and MetALD were identified as significant independent risk factors for PH-PVT. PH-PVT incidence was lower in patients with other etiologies of SLD or metabolic dysfunction without SLD, suggesting that hepatic steatosis and metabolic dysfunction together play a synergistic role in the development of PVT. Careful postoperative attention is necessary, including early postoperative examination with enhanced CT or other imaging methods in patients with MASLD and MetALD. Anticoagulation therapy is recommended immediately after the identification of PVT. Although further studies are needed, the results of this study offer promising suggestions for surgeons and hepatologists.

## Figures and Tables

**Figure 1 cancers-16-03844-f001:**
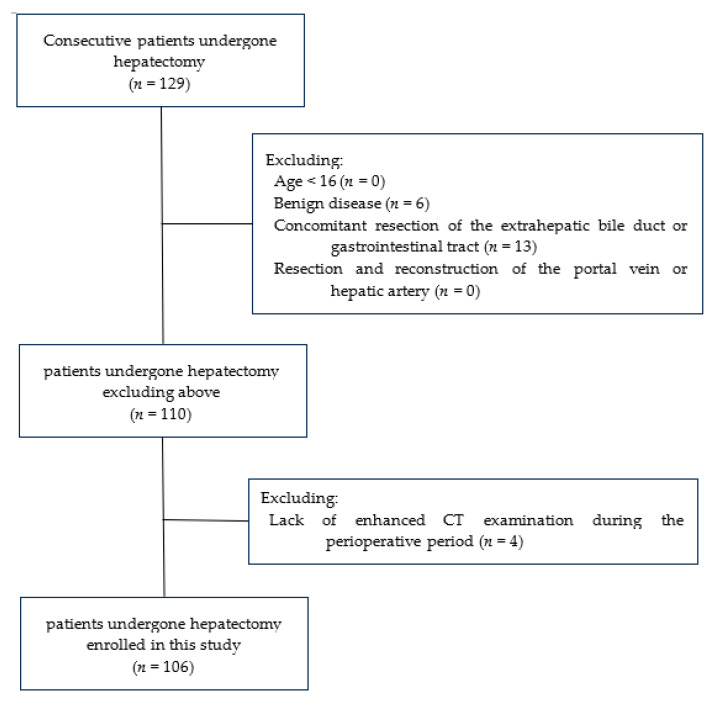
Flow chart of patient selection.

**Figure 2 cancers-16-03844-f002:**
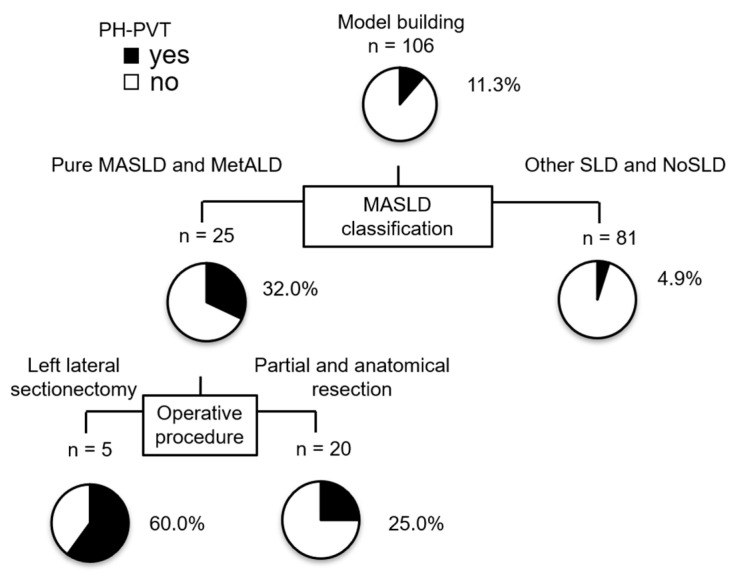
Profiles associated with hepatectomy. Decision-tree algorithm for post-hepatectomy portal vein thrombosis (PH-PVT). The pie graphs indicate the percentage of PH-PVT (black)/absence (white) in each group.

**Table 1 cancers-16-03844-t001:** Patient characteristics.

		Value or *n* (%)	Range
Age, years old		70.2 ± 9.1	42–90
Sex Female/Male		25/81	
Diagnosis of liver tumor			
Hepatocellular carcinoma		50 (47.2)	
Intrahepatic cholangiocarcinoma		7 (6.6)	
Intrahepatic cholangiolocarcinoma		2 (1.9)	
Combined hepatocellular-cholangiocarcinoma		1 (0.9)	
Metastatic liver cancer		46 (43.4)	
SLD category			
MASLD		20 (18.9)	
MetALD		5 (4.7)	
Other SLD		13 (12.3)	
No SLD		68 (64.2)	
ALT, U/L		24.6 ± 13.7	6–76
Total bilirubin, mg/dL		0.63 ± 0.27	0.3–1.3
Albumin, g/dL		4.0 ± 0.7	3.3–4.6
Prothrombin time, %		96.3 ± 13.1	75.3–122.7
Platelet count, ×10^4^		18.6 ± 6.4	7.0–45.4
Child-Pugh score (point) *	5/6/7 or more	94(90.4)/10(9.6)/0(0)	
Tumor size, mm		32.1 ± 24.6	9–120
Tumor number	single/multiple	80 (75.5)/26 (24.5)	
Vascular tumor thrombus	yes/no	3 (2.8)/103 (97.2)	
Previous chemotherapy **	yes/no	24 (22.6)/82 (77.4)	
Previous anticoagulant administration ***	yes/no	20 (18.9)/86 (81.1)	
PVE prior to surgery	yes/no	10 (9.4)/96 (90.6)	

Values are presented as a mean ± standard deviation or number. * Because the preoperative prothrombin time was unreliable due to warfarin administration not being ceased at the time of measurement, the Child-Pugh score (point) was not available in two patients. ** Previous chemotherapy: chemotherapy received within six months before the hepatectomy. *** Oral anticoagulant therapy owing to comorbidities such as arrhythmia, cardiac disease, and cerebrovascular disease.

**Table 2 cancers-16-03844-t002:** Operative and postoperative course.

		Value or *n* (%)	Range
Type of hepatectomy	partial resection	40 (37.7)	
	left lateral sectionectomy	15 (14.2)	
	anatomical resection	51 (48.1)	
Laparoscopic surgery	yes/no	29 (27.4)/77 (72.6)	
Concomitant RFA	yes/no	9 (8.5)/97 (91.5)	
Operation time (min)		332.7 ± 126.0	127–600
Intraoperative Blood loss (mL)		309.1 ± 336.4	0–1510
Blood transfusion	yes/no	7 (6.6)/99 (93.4)	
Pringle’s maneuver (times × 15 min)		4.0 ± 2.5	0–12
Postoperative venous thrombosis *			
PVT		11 (10.4)	
PVT + HVT + VCT		1 (0.9)	
PTE		1 (0.9)	
DVT		1 (0.9)	
Morbidity excluding PVT (Clavien-Dindo grade)			
None		74 (69.8)	
I or II		7 (6.6)	
III		22 (20.8)	
IV		3 (2.8)	
V		0 (0)	
Mortality within 90 days after surgery		0 (0)	

* Venous thrombotic complication in the postoperative acute period. PVT; portal vein thrombosis, HVT; hepatic vein thrombosis, VCT; vena cava thrombosis, PTE; pulmonary thrombotic embolization, DVT; deep vein thrombosis.

**Table 3 cancers-16-03844-t003:** Location and stenosis rate of post-hepatectomy portal vein thrombosis.

Post-Hepatectomy Portal Vein Thrombosis (PH-PVT)	*n* (%)
Location	
The third branch of the portal vein	1 (8.3)
The second branch of the portal vein—The umbilical portal vein	8 (66.7)
Other portal veins	0 (0)
Right or left portal vein	3 (25.0)
Main portal vein or superior mesenteric vein	0 (0)
Stenotic rate	
Less than 50%	2 (16.7)
50~70%	4 (33.3)
70~90%	3 (25.0)
90~100%	3 (25.0)

**Table 4 cancers-16-03844-t004:** Univariate factor analysis for post-hepatectomy portal vein thrombosis.

	Post-Hepatectomy Portal Vein Thrombosis (PH-PVT)		
	No (%)	Yes (%)	*p* Value
n	94 (88.7)	12 (11.3)	
Age	70.1 ± 9.1	71.6 ± 9.4	0.5829
Sex female male	21 (84.0) 73 (90.1	4 (16.0) 8 (9.9)	0.3983
Disease PLC MLC	52 (86.7) 42 (91.3)	8 (13.3) 4 (8.7)	0.4551
SLD category			0.0019
MASLD	13 (65.0)	7 (35.0)	
MetALD	4 (80.0)	1 (20.0)	
Other SLD	12 (92.3)	1 (7.7)	
No SLD	65 (95.6)	3 (4.4)	
Total bilirubin, mg/dL	0.63 ± 0.26	0.66 ± 0.28	0.50155
ALT, IU/L	24.7 ± 13.0	23.6 ± 19.0	0.87174
Albumin, g/dL	4.0 ± 0.4	4.0 ± 0.3	0.74719
Prothrombin time, %	96.1 ± 13.4	97.3 ± 10.6	0.97771
Platelet count, ×10^4^	18.5 ± 6.6	19.1 ± 5.1	0.84626
Child-Pugh score (point) 5 6 7 or more	83 (88.3) 9 (90.0) 0 (0)	11 (11.7) 1 (10.0) 0 (0)	0.8727
Tumor size, mm	31.8 ± 24.5	34.2 ± 26.3	0.7592
Tumor number Single **Multiple**	69 (86.3) 25 (96.2)	11 (13.8) 1 (3.9)	0.1662
Macroscopic vascular invasion			0.5301
Absent	91 (88.4)	12 (11.7)	
Present	3 (100)	0	
Previous chemotherapy *			0.2085
Yes	23 (95.8)	1 (4.2)	
No	71 (86.6)	11 (13.4)	
Previous anticoagulant **			0.1738
Yes	16 (80.0)	4 (20.0)	
No	78 (90.7)	8 (9.3)	
Preoperative Portal vein embolization			0.2351
Yes	10 (100)	0	
No	84 (87.5)	12 (12.5)	
Type of hepatectomy			0.0057
Partial resection	39 (97.5)	1 (2.5)	
Anatomical resection	45 (88.2)	6 (11.8)	
Left lateral sectionectomy	10 (66.7)	5 (33.3)	
Operative approach			0.8457
Laparoscopic	26 (89.7)	3 (10.3)	
Laparotomy	68 (88.3)	9 (11.7)	
Operation time, min	335 ± 126	307 ± 127	0.4651
Intraoperative bleeding, mL	320 ± 337	224 ± 333	0.3552
Pringle maneuver, times × 15 min.	3.9 ± 2.4	4.5 ± 3.3	0.4979
Blood transfusion			0.3280
**Yes**	7 (100)	0	
**No**	87 (87.9)	12 (12.1)	
Concomitant RFA			0.2625
**Yes**	9 (100)	0	
**No**	85 (87.6)	12 (12.4)	
Postoperative complications excluding PH-PVT			0.9551
None	66 (89.2)	8 (10.8)	
Minor (Clavien-Dindo I, II)	6 (85.7)	1 (14.3)	
Major (Clavien-Dindo III, IV, V)	22 (88.0)	3 (12.0)	

Values are presented as a mean ± standard deviation or number. PLC, primary liver cancer; MLC, metastatic liver cancer; Other SLD includes ALD, specific etiology SLD, and cryptogenic SLD. No SLD represents liver without steatosis. * Previous chemotherapy: chemotherapy received within six months prior to hepatectomy. ** Previous anticoagulant: oral anticoagulant therapy owing to comorbidities such as arrhythmia, cardiac disease, and cerebrovascular disease.

**Table 5 cancers-16-03844-t005:** Multivariate analysis of risk factors for post-hepatectomy portal vein thrombosis.

	Odds Ratio	95% CI	*p* Value
SLD category (MASLD and MetALD)	9.27	2.32–37.08	0.0016
Type of hepatectomy Left lateral sectionectomy	6.22	1.40–27.67	0.0165

**Table 6 cancers-16-03844-t006:** Univariate analysis of the incidence of post-hepatectomy portal vein thrombosis in MASLD and No SLD, along with metabolic factors, excluding alcohol consumption.

	Post-Hepatectomy Portal Vein Thrombosis (PH-PVT)		
	No (%)	Yes (%)	*p* Value
*n*	51 (86.4)	8 (13.6)	
MASLD No SLD along with metabolic factors, excluding alcohol consumption	13 (65.0) 38 (97.4)	7 (35.0) 1 (2.6)	0.0006

## Data Availability

Data that support the findings of this study are available from the author, K.O. (Koji Okuda), upon reasonable request.

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
