# Peer review of "Impact of MASLD on Portal Vein Thrombosis Following Hepatectomy for Liver Cancer"

_cancers, 2024, doi:10.3390/cancers16223844_

Round 1

Reviewer 1 Report

Comments and Suggestions for Authors

Dear Author, I give you the following comment. Please address this in your manuscript to enhance the readability and understanding of your manuscript.

Major Comments:

  1. Study Design and Population:
    • How were the patients selected for inclusion in the study? Was there any potential selection bias in choosing the 106 patients who underwent hepatectomy for liver cancer?
    • Were there any standardized criteria or guidelines used to classify liver steatosis into MASLD, MetALD, Other SLD, and No SLD? Were these classifications consistently applied across all patients?
  2. Multivariate Analysis:
    • Could you clarify the selection of variables for the multivariate analysis? How were potential confounders controlled to ensure that MASLD was an independent risk factor for PH-PVT?
    • Did the study account for other potential variables contributing to PH-PVT development post-hepatectomy, such as the extent of liver resection or pre-existing liver conditions?
  3. Alcohol Consumption:
    • Can you elaborate on how alcohol consumption was measured and classified in the context of MetALD? Were there specific thresholds of alcohol intake to distinguish between MASLD and MetALD?
  4. Statistical Power:
    • The MASLD and MetALD groups had relatively small sample sizes (n=20 and n=5, respectively). How was the study's statistical power ensured, particularly for detecting significant differences between these small groups and the No SLD group?
  5. Clinical Implications and Recommendations:
    • What are the clinical recommendations based on your findings? Should patients with MASLD and MetALD be treated or monitored differently when undergoing hepatectomy? Could pre-operative interventions aimed at improving metabolic function reduce the risk of PH-PVT?

6.      Schematics: Please Add schematics in an introduction section to highlight your works.

Minor Comments:

  1. Definition of Terms:
    • In the text, it may be helpful to more clearly define the term "PH-PVT" in its first usage for readers who may not be familiar with the abbreviation.
  2. Statistical Methods:
    • Were the specific statistical methods used (e.g., logistic regression, chi-square tests) reported in the methods section? Including these details would help readers understand how the results were derived.
  3. Grammar and Terminology:
    • There are a few grammatical inconsistencies, such as the phrase "with metabolic factors but alcohol consumption" in the abstract, which seems unclear. Revising this could improve readability.
  4. Data Presentation:
    • The results on PH-PVT incidences across the different groups could be presented more clearly, perhaps in a table format to visually differentiate the percentages and group classifications.
  5. References to Literature:
    • Are there any references to support the idea that MASLD and MetALD play a synergistic role in the development of PH-PVT? Including more literature citations could strengthen this claim.

These questions aim to address both overarching concerns and specific technical details that could impact the robustness and clarity of the study's findings.

Best Regards

Comments on the Quality of English Language

English is fine.

Author Response

To REVIEWER 1

Thank you very much for your letter regarding our manuscript (Cancers-23275601). We appreciate your comments, which have helped us to improve our manuscript. In line with your comments, please find below our point-by-point responses.

Major

Study design and population

  1. How were the patients selected for inclusion in the study? Was there any potential selection bias in choosing the 106 patients who underwent hepatectomy for liver cancer?

Response: We indeed appreciate your comments and apologize for our insufficient description in the manuscript. In this retrospective study, 129 consecutive patients who underwent hepatectomy between June 2011 and December 2023 at Kyoaikai Tobata Kyoritsu Hospital were initially enrolled. The following exclusion criteria were applied: 1) patients aged less than 16 years (n=0); 2) those who had a hepatectomy for benign diseases (n=6); 3) patients with concomitant resection of the extrahepatic bile duct or gastrointestinal tract (n=13); 4) patients who required resection and reconstruction of the portal vein or hepatic artery (n=0); and 5) patients lacking enhanced CT imaging during the perioperative period to identify portal vein thrombosis (n=4). In this study, we excluded benign tumors. It is generally recognized that patients with malignant tumors are more likely to have a hypercoagulable state caused by tumor-derived cytokines, circulating mucin molecules, plasminogen activator inhibitor-1 inducing tumor hypoxia, etc. [3]. There may be different risks for postoperative venous thrombosis between benign and malignant tumors. Thus, we excluded benign tumors in this study. We also exclude patients who are unable to undergo enhanced CT scans due to renal insufficiency or allergic reactions to the CT contrast medium. This is because diagnosing the presence or absence of PVT in the early postoperative phase is difficult without the findings from enhanced CT imaging. Ultimately, 106 patients were included in the analysis. In this process, there are no potential biases in the selection of the patients for analysis. We added an explanation (lines 79-85) and the figure titled "Flow Chart of Patient Selection" in section 2.1.

  1. Were there any standardized criteria or guidelines used to classify liver steatosis into MASLD, MetALD, Other SLD, and No SLD? Were these classifications consistently applied across all patients?

Response: We apologize for the lack of sufficient information. The classification and definition of steatotic liver disease, which includes MASLD and MetALD, are detailed in reference number 4 in the "References" section [1]. Furthermore, the classification has received support and endorsement from various societies and organizations worldwide. The diagnostic procedures can be found on the websites of organizations such as the American Association for the Study of Liver Disease (AASLD), the Latin American Association for the Study of the Liver (ALEH), and the European Association for the Study of the Liver (EASL) [2]. This reference is cited as reference number 5 in the "References" section. (Line 138)

Multivariate Analysis:

  1. Could you clarify the selection of variables for the multivariate analysis? How were potential confounders controlled to ensure that MASLD was an independent risk factor for PH-PVT?

Response: We apologize for not providing enough information for the multivariate analysis. To investigate independent factors associated with PH-PVT, explanatory variables were selected using a stepwise manner minimizing the Bayesian information criterion as previously described [3, 4]. The following factors were applied in the stepwise procedure: age, sex, disease, steatotic liver disease category, total bilirubin, ALT, albumin, prothrombin time, platelet count, Child-Pugh score, tumor size, tumor number, macroscopic vascular invasion, previous chemotherapy, previous anticoagulant, preoperative portal vein embolization, type of hepatectomy, operative approach, operation time, intraoperative bleeding, Pringle maneuver, blood transfusion, concomitant RFA and postoperative complications. The selected variables were used in a logistic regression analysis to identify independent factors for PH-PVT. We added more detailed information in section 2.6 (lines 204-211).

  1. Did the study account for other potential variables contributing to PH-PVT development post-hepatectomy, such as the extent of liver resection or pre-existing liver conditions?

Response: We appreciate the valuable comments provided. Metabolic disorders, such as obesity and type 2 diabetes mellitus, along with NAFLD, contribute to an inflammatory environment that increases the risk of venous thrombosis. As a result, we hypothesized that metabolic dysfunction-associated steatotic liver disease; MASLD, is a risk factor for PH-PVT. Additionally, previous studies have identified several factors that may elevate the risk of PH-PVT. These include the duration of operation, intraoperative bleeding volume, the length of the Pringle maneuver, and anatomical resections—specifically, the combined resection of portal segments, and a particular emphasis on left lateral sectionectomy [5]. Furthermore, it has been suggested that portal vein embolization prior to liver resection, and the administration of oral anticoagulants due to comorbidities such as arrhythmia, cardiac disease, and cerebrovascular disease (those were discontinued perioperative period in a usual manner), as well as prior chemotherapy that may lead to sinusoidal injury or steatohepatitis, could also be risk factors for PH-PVT. These explanatory variables were analyzed using a stepwise approach. In section 2.6, we have provided a description of the variables used in the stepwise procedures (lines 204-210).

Alcohol Consumption:

  1. Can you elaborate on how alcohol consumption was measured and classified in the context of MetALD? Were there specific thresholds of alcohol intake to distinguish between MASLD and MetALD?

Response: I aporozize for our insufficient information regarding alcohol consumption. At our institution, we conducted interviews regarding alcohol consumption at the time of admission for hepatectomy. The details and quantity of alcohol consumed were recorded. In this study, we utilized the alcohol conversion table proposed by the Japanese Ministry of Health, Labour and Welfare. The threshold of alcohol consumption distinguishing MASLD from MetALD was outlined by Rinella, M.E., and colleagues in their multi-society Delphi consensus statement on the new nomenclature for fatty liver disease, published in J Hepatol 2023, volume 79, pages 1542-1556, specifically in Figure 5 [1] (see Table) (listed as number 4 in the “References” section). We added this explanation in the section 2.2. (lines 149-151).

Table. Specific thresholds of alcohol intake for MetALD

Gender

Daily alcohol intake

Weekly alcohol intake

Female

20–50 g

140–350 g

Male

30–60 g

210–420 

Statistical Power

  1. The MASLD and MetALD groups had relatively small sample sizes (n=20 and n=5, respectively). How was the study's statistical power ensured, particularly for detecting significant differences between these small groups and the No SLD group?

Response: As you pointed out, we totally agreed that the sample size of MASLD and MetALD was small, and we were not able to increase the sample size immediately. Therefore, we added this issue as a limitation (lines 389-392). However, the following may be possible reasons for the statistically significant differences obtained in this study. MASLD and MetALD groups were relatively homogeneous groups because of their inclusion criteria. By targeting a population with low variability, we might be able to detect effects with a small sample size. In addition, for a large sample size group (No SLD group), we properly handled missing values and outliers to maintain data consistency and improve analysis accuracy. Thus, statistical power might be ensured in this study.

Clinical Implications and Recommendations:

  1. What are the clinical recommendations based on your findings? Should patients with MASLD and MetALD be treated or monitored differently when undergoing hepatectomy? Could pre-operative interventions aimed at improving metabolic function reduce the risk of PH-PVT?

Response: Thank you for your valuable comments. PH-PVT is a potentially life-threatening complication and is significantly prevalent in patients with MASLD and MetALD. Careful postoperative attention is necessary, such as early postoperative examination of enhanced CT or ultrasonography in patients with MASLD and MetALD. Anticoagulation therapy is recommended immediately after the identification of PVT. We added clinical recommendations for patients with MASLD and MetALD who are undergoing hepatectomy (lines 406-409). We currently have limited evidence on preoperative interventions aimed at enhancing metabolic function and reducing the risk of PH-PVT. Additionally, early postoperative prophylactic administration of anticoagulants or AT III supplementation may reduce the likelihood of PH-PVT development; however, further studies are required.

  1. Schematics: Please Add schematics in an introduction section to highlight your works.

Response: Thank you for your kind suggestion. We added the figure titled “Flow chart of patient selection” in section 2.1.       

Minor Comments

Definition of Terms:

  1. In the text, it may be helpful to more clearly define the term "PH-PVT" in its first usage for readers who may not be familiar with the abbreviation.

Response: Thank you for your kind comment. As per your suggestion, we revised it as follows: ”portal vein thrombosis following hepatectomy (PH-PVT)” has been changed to “post-hepatectomy portal vein thrombosis (PH-PVT)” (lines 13, 29, 52)

Statistical Methods:

  1. Were the specific statistical methods used (e.g., logistic regression, chi-square tests) reported in the methods section? Including these details would help readers understand how the results were derived.

Response: We appreciate your feedback and apologize for not adequately explaining the statistical analysis. We have included additional details about the statistical method in section 2.6. “Statistical analysis”.

Grammar and Terminology:

  1. There are a few grammatical inconsistencies, such as the phrase "with metabolic factors but alcohol consumption" in the abstract, which seems unclear. Revising this could improve readability.

Response: We apologize for any errors in our English grammar. To improve our writing, we have had our text proofread again by Editage (www.editage.com). The phrase "with metabolic factors but alcohol consumption" has been revised to "in patients without SLD, along with metabolic factors, excluding alcohol consumption."

A proofreading certificate is attached.

Data Presentation:

  1. The results on PH-PVT incidences across the different groups could be presented more clearly, perhaps in a table format to visually differentiate the percentages and group classifications.

Response: Thank you for your kind instruction. We modified the format of Table 4 to enhance its visual clarity.

References to Literature:

  1. Are there any references to support the idea that MASLD and MetALD play a synergistic role in the development of PH-PVT? Including more literature citations could strengthen this claim.

Response: Thank you for your valuable comment. NAFLD is closely linked to insulin resistance and is commonly observed in patients with metabolic dysfunction, such as obesity and type 2 diabetes mellitus. It is well established that both obesity and NAFLD contribute to chronic inflammation and create a significant procoagulant state, which can ultimately lead to thrombosis [6-9]. Interestingly, our study found that patients with steatotic liver disease without metabolic dysfunction, as well as those with metabolic dysfunction without steatotic liver disease, were not at risk for PH-PVT. In contrast, patients with MASLD and MetALD were at significantly higher risk. This suggests a synergistic effect of hepatic steatosis and metabolic dysfunction in the onset of PH-PVT. Unfortunately, we have not found any literature to support this phenomenon, either physiologically or clinically, and further studies are necessary to validate our hypothesis.

Reference

  1. Rinella M.E, et al. A multisociety Delphi consensus statement on new fatty liver disease nomenclature. J Hepatol 2023;79:1542-1556
  2. “AASLD's MASLD Decision Tree“ (SLD-597 AASLD 2023 MASLD Decision Tree Digital (sm)1.8.pdf )
  3. Fukunaga S, et al. Impact of non-obese metabolic dysfunction-associated fatty liver disease on risk factors for the recurrence of esophageal squamous cell carcinoma treated with endoscopic submucosal dissection: A multicenter study. Hepatol Res. 2024;54:201-212.
  4. Sano T, et al. Metabolic management after sustained virologic response in elderly patients with hepatitis C virus: A multicenter study. Hepatol Res. 2024;54:326-335.
  5. Mori A et al. Risk Factors and Outcome of Portal Vein Thrombosis After Laparoscopic and Open Hepatectomy for Primary Liver Cancer: A Single-Center Experience. World J Surg 2020;44:3093-3099.
  6. Morange PE, et al. Thrombosis in central obesity and metabolic syndrome: mechanisms and  Thromb Haemost 2013;110:669-680.
  7. Alessi MC, et al. PAI-l and the metabolic syndrome: links, causes, and consequences. Arterioscler Thromb Vasc Biol 2006;10:2200-2207.
  8. Stine JG, et al. Increased risk of venous thromboembolism in hospitalized patients with cirrhosis due to non-alcoholic steatohepatitis. Clin Transl Gastroenterol 2018;9:140.
  9. Tripodi A, et al. Procoagulant imbalance in patients with non-alcoholic fatty liver disease. J Hepatol 2014;61:148-154.

Reviewer 2 Report

Comments and Suggestions for Authors

This study investigates the impact of MASLD on postoperative portal vein thrombosis after hepatectomy. I have a few remarks:

Methods:

1.     The authors report on 106 patients over a course of 12 years. This inherits a lot of confounders incl. surgeons, techniques, indications.

2.     Why were benign diseases excluded?

3.     Why did patients receive CTs after surgery? The authors state that 4 patients did not receive imaging and were therefore excluded; however, one could assume that no thrombosis was present/clinical significant to lead to such diagnosis in these cases?

4.     The definition of hepatectomy is questionable:

As is understand, partial resection means atypical/wedge resection? But the other two classifications are anatomical and LLS. However, prior, the author stated that “Individual ligation of the hepatic vessels was used in hemihepatectomy, extended hemihepatectomy, and trisectionectomies”. That means these type of surgery were all included in “anatomical resection”? The incidence of complications in major hepatectomy incl. pv thrombosis differs quite a lot between minor pure anatomical resection of one segment vs e.g. trsectionectomy.

Results:

1.     Table 1: What was the difference between Intrahepatic cholangiocarcinoma n=7 and n=2?

2.     Please add percentages, when n is given

3.     Again, regression analysis using “left lateral sectionectomy] vs. [partial and anatomical resection]” is very questionable. And why were the latter two combined? I would suspect, that major hepatectomy has one of the highest risk of thrombosis.

4.     Table 3 How many patients did not receive prophylactic anticoagulation? Why were patients with therapeutic anticoagulation discontiniued before surgery and when was therapy reinitiated? Again, surgical resection extent should be assessed differently.

5.     Table 3: What does Child-Pugh Score 5 and 6 mean?

6.     Table: I would suggest to dichotomize a few variables (e.g. operation time <120 min vs >120min, Cheng et al 2018).

7.     Table 3: “No” can be left out )Blood transfusion, RFA)

8.     Why did 2 patients receive no treatment of thrombosis?

9.     Where was thrombus located exactly and how many pat. had complete occlusion?

10.  “In all patients, PH-PVT completely disappeared within 6 months after hepatectomy without invasive surgical thrombectomy or thrombolytic treatment via a catheter positioned in the superior mesenteric artery or portal vein.” This sentence can be shortened to “In all patients, PH-PVT completely disappeared within 6 months after hepatectomy without invasive thrombectomy”

11.  As far as I can see, “postoperaticve complications” were not assessed with their impact on thrombosis? Why? Complication after surgery often lead to extended hospital stay, re-intervtions and increased immobilization, that increase risk of thrombosis.

12.  Extent of steatosis certainly might be relevant. This was not assessed?

Discussion and Introduction;

1.     I would recommend to stick to definition of e.g. MASLD and not switch between NAFLD and MASLD. It has no merit here to mention the switch/new definition.

Author Response

To REVIEWER 2

Thank you very much for your letter regarding our manuscript (Cancers-23275601). We appreciate your comments, which have helped us to improve our manuscript. In line with your comments, please find below our point-by-point responses.

Methods:

  1. The authors report on 106 patients over a course of 12 years. This inherits a lot of confounders incl. surgeons, techniques, indications.

Response: We appreciate your valuable comment. In this study, the explanatory variables were selected using a stepwise manner minimizing the Bayesian information criterion as previously described [1,2]. The following factors were applied in the stepwise procedure: age, sex, disease, steatotic liver disease category, total bilirubin, ALT, albumin, prothrombin time, platelet count, Child-Pugh score, tumor size, tumor number, macroscopic vascular invasion, previous chemotherapy, previous anticoagulant, preoperative portal vein embolization, type of hepatectomy, operative approach, operation time, intraoperative bleeding, Pringle maneuver, blood transfusion, and concomitant RFA. The selected variables were used in a logistic regression analysis to identify independent factors for PH-PVT. We added how potential confounders were controlled to ensure an independent risk factor for PH-PVT in section 2.6. (lines 202-211). We properly handled outliers to maintain data consistency and improve analysis accuracy. Thus, statistical power might be ensured in this study. However, as you indicated, the results should be reconfirmed in further large-sample studies. Therefore, we added this issue as a limitation (lines 385-392).

  1. Why were benign diseases excluded?

Response: We thank you for your valuable indication. We apologize that we did not provide information on why we excluded the benign disease in the manuscript. It is generally recognized that patients with malignant tumors are more likely to have a hypercoagulable state caused by tumor-derived cytokines, circulating mucin molecules, plasminogen activator inhibitor-1 inducing tumor hypoxia, etc. [3]. There may be different risks for postoperative venous thrombosis between benign and malignant tumors. Thus, we excluded benign tumors in this study. We added the reason for excluding the patients with benign tumors in section 2.1. (lines 79-82).

  1. Why did patients receive CTs after surgery? The authors state that 4 patients did not receive imaging and were therefore excluded; however, one could assume that no thrombosis was present/clinical significant to lead to such diagnosis in these cases?

Response: We apologize for not providing information regarding postoperative CT examinations. At our institution, we routinely perform enhanced CT scans to detect postoperative complications—such as intraperitoneal abscesses, biloma, hepatic infarction, portal vein thrombosis, and others—between 4 and 7 days after surgery. Additionally, we conduct CT examinations at 3 and 6 months postoperatively as part of our routine follow-up care. We excluded 4 patients who could not undergo enhanced CT due to renal insufficiency or allergic reactions to the iodine contrast medium. While you may suggest that PVT could be diagnosed based on clinical manifestations, it is often asymptomatic following a hepatectomy. When it progresses, PVT can lead to severe liver damage due to reduced portal flow and portal hypertension. Thus, diagnosing the presence or absence of PVT in the early postoperative phase is challenging without enhanced CT findings. Consequently, we excluded patients who did not undergo postoperative enhanced CT. This explanation has been added to section 2.1 (lines 82-85).

  1. The definition of hepatectomy is questionable: As is understand, partial resection means atypical/wedge resection? But the other two classifications are anatomical and LLS. However, prior, the author stated that “Individual ligation of the hepatic vessels was used in hemihepatectomy, extended hemihepatectomy, and trisectionectomies”. That means these type of surgery were all included in “anatomical resection”? The incidence of complications in major hepatectomy incl. pv thrombosis differs quite a lot between minor pure anatomical resection of one segment vs e.g. trsectionectomy.

Response: We appreciate your valuable comments and apologize for not providing detailed information about the types of hepatectomy. Specifically, we classified the type of resection into “partial resection”, “left lateral sectionectomy” and “anatomical resection”. We defined "partial resection" to include atypical resections and wedge resection, while "anatomical resection" refers to systematic resections of Couinaud segment, section, the right and left hemiliver, and the right and left trisection, following the Brisbane 2000 system of nomenclature for hepatic anatomy and resections [4]. However, we excluded left lateral sectionectomy from the category of "anatomical resection" in this study, placing it in an independent category instead. These definitions have been included in section 2.6. (lines 165-172). This distinction was made for the following reasons. We fully agree with your comment that the risks, including portal vein thrombosis, may differ between minor anatomical resections of one segment or section and more extensive procedures, such as bisectionectomy and trisectionectomy. Previous research indicated that combined portal segment resections, especially right hemihepatectomies, can be risk factors for PH-PVT [5, 6]. However, in our study data, the incidence of PH-PVT was not high for resection of bisection and trisection procedures, including hemihepatectomy, central bisectionectomy, and trisectionectomy. In contrast, minor anatomical resections, such as mono-segmentectomy and mono-sectionectomy, had a relatively higher incidence (see Table). These findings are inconsistent with previously published results. We need further studies with large sample sizes and added this issue as a limitation (lines 386-389, 391-392).

Our data also indicated that the highest incidence was observed in left lateral sectionectomy, which is consistent with findings from several other studies. As discussed in the “Discussion” section, there may be certain factors associated with left lateral sectionectomy that induce the development of PH-PVT. Consequently, we categorized the types of resection into three groups: "partial resection," "left lateral sectionectomy," and "anatomical resection excluding left lateral sectionectomy."     

Table incidence of PH-PVT by type of hepatectomy

Resection of

No.

PH-PVT (%)

Partial

40

1 (2.5)

Left lateral section (LLS)

15

5 (33.3)

Mono-segment or mono-section excluding LLS

20

5 (25.0)

Bisection

29

1 (3.4)

Trisection

2

0 (0)

Results:

  1. Table 1: What was the difference between Intrahepatic cholangiocarcinoma n=7 and n=2?

Response: Thank you for your comment. In Table 1, we described: "Intrahepatic cholangiocarcinoma (n=7)" and "Intrahepatic cholangiolocarcinoma (n=2)." Cholangiolocarcinoma (CLC) is a variant of intrahepatic cholangiocarcinoma (iCCA) and is defined by more than 80% of the tumor area exhibiting cholangiolocellular differentiation without any hepatocellular differentiation. CLC is thought to originate from the bile ductule, which contains hepatic stem or progenitor cells and the canals of Hering. Previously, CLC was classified as a subtype of combined hepatocellular-cholangiocarcinoma. However, recent molecular profiling studies now indicate that it should be classified as part of iCCA [7]. The current World Health Organization (WHO) classification recognizes CLC as a subtype of small bile duct intrahepatic cholangiocarcinoma.

  1. Please add percentages, when n is given

Response: Thank you for your thoughtful suggestion. We included the percentage where the n has been provided in Tables 1, 2, and 3.

  1. Again, regression analysis using “left lateral sectionectomy] vs. [partial and anatomical resection]” is very questionable. And why were the latter two combined? I would suspect, that major hepatectomy has one of the highest risk of thrombosis.

Response: We apologize for not providing sufficient information for the multivariate analysis. The type of resection was classified as "partial resection," "left lateral sectionectomy," and "anatomical resection," as explained above (Comment response to “Method 4”). To investigate independent factors associated with PH-PVT, explanatory variables were selected using a stepwise manner minimizing the Bayesian information criterion as previously described [1,2]. Several factors were applied to stepwise procedure and selected variables were used in a logistic regression analysis to identify independent factors for PH-PVT. In the stepwise procedure, regarding the type of resection, “left lateral sectionectomy” vs. “partial resection + anatomical resection” was selected as the only significant combination for the risk of portal vein thrombosis (p=0.014). Therefore, we stratified into “left lateral sectionectomy” and “partial resection + anatomical resection” for a logistic regression analysis. As shown in the table in the comment response “Method 4”, our data found that the incidence of PH-PVT was not high for bisection and trisection procedures, including hemihepatectomy, central bisectionectomy, and trisectionectomy, compared to other type of resections. As you pointed out, this finding was inconsistent with prior research. Therefore, further studies with larger sample sizes are necessary, and we have noted this limitation in “Discussion”.

  1. Table 3 How many patients did not receive prophylactic anticoagulation? Why were patients with therapeutic anticoagulation discontiniued before surgery and when was therapy reinitiated? Again, surgical resection extent should be assessed differently.

Response: We sincerely apologize for our unclear explanation regarding perioperative anticoagulant therapy. Twenty patients were used to be on oral anticoagulant therapy due to comorbidities such as arrhythmia, cardiac disease, and cerebrovascular disease. These oral anticoagulants were discontinued before surgery in a usual manner to avoid intraoperative bleeding tendency. For these patients, anticoagulant therapy was resumed at 3 to 5 days postoperatively. The other 86 patients did not receive any preoperative and postoperative prophylactic anticoagulant therapy to prevent venous thrombosis. When PH-PVT was identified, anticoagulation therapy was promptly initiated with systemic heparinization, with or without AT III supplementation, followed by oral warfarin or direct oral anticoagulants (DOAC). Few patients were treated with warfarin or DOAC alone. We add this explanation in the revised manuscript (lines 226-231).

Regarding the extent of resection, we described it above. We kindly ask you to consider our viewpoint on this issue.

  1. Table 3: What does Child-Pugh Score 5 and 6 mean?

Response: We sincerely apologize for the confusion surrounding the Child-Pugh scoring notation. Generally, the Child-Pugh grading system is classified into three categories: grade A (5-6 points), grade B (7-9 points), and grade C (10-15 points). In our study, when we referred to the Child-Pugh score, we specifically described these point values. It is important to note that all patients in our study had a Child-Pugh grade A, and therefore, we are only using the points. For clarity, we have revised the Table 1 notation accordingly.

  1. Table: I would suggest to dichotomize a few variables (e.g. operation time <120 min vs >120min, Cheng et al 2018).

Response: Thank you for your valuable comment. We agree that dichotomizing continuous variables is an important aspect of statistical analysis. However, in this study, we selected explanatory variables using a stepwise method for logistic regression analysis, as we mentioned earlier (Comment response “Method 1”). In this stepwise procedure, continuous variables can be directory applied to identify factors for logistic regression analysis.

  1. Table 3: “No” can be left out )Blood transfusion, RFA)

Response: We apologize for our indistinct Table 3, which has been revised to Table 4. We would like to provide information on the presence or absence of PH-PVT and the corresponding percentage for both the "No" and "Yes" categories, including those for blood transfusion and RFA. Instead of it, we modified the table format of Table 4 to enhance its visual clarity.

  1. Why did 2 patients receive no treatment of thrombosis?

Response: We sincerely apologize for our unclear explanation. Anticoagulant therapy was not administered to two patients with PH-PVT. One patient had an anaphylactic reaction to anticoagulants, while in the other patient, the development of PH-PVT went unnoticed during the postoperative period. However, PH-PVT was later identified through a retrospective examination of CT images for this study. This explanation had been described in the discussion section; however, it would be better to present it in the results section. We referenced this explanation in section 3.4. (lines 304-307).

  1. Where was thrombus located exactly and how many pat. had complete occlusion?

Response: We appreciate your valuable comments and apologize for the insufficient description provided. As we had described in section 3.2, it was not easy to understand the details. Thus, we have added a new table titled “Location and stenosis rate of post-hepatectomy portal vein thrombosis (PH-PVT)” as Table 3 and updated the description in the manuscript. (lines 248-253). PH-PVT was identified in the first branch of the left or right portal vein in three patients, in the umbilical portal vein in eight patients, and in the third branch of the left lobe in one patient. None of the patients had thrombus present in the main portal vein or to a greater extent.

The stenosis rates were categorized as follows: less than 50% in two patients, between 50% to 70% in four patients, 70% to 90% in three patients, and over 90% in three patients. Among the last group, one patient experienced complete obstruction caused by PH-PVT located in the umbilical portal vein, which occurred after a left lateral sectionectomy.

Table.  Location and stenosis rate of post-hepatectomy portal vein thrombosis (PH-PVT)

Post-hepatectomy portal vein thrombosis

n (%)

Location

   Third branch of the portal vein

   Second branch of the portal vein- Umbilical portal vein

-       Others  

   Right or left portal vein

   Main portal vein or superior mesenteric vein

1 (8.3)

8 (66.7)

0 (0)

3 (25.0)

0 (0)

Stenotic rate

   Less than 50 %

   50~70 %

   70 ~90 %

   90~100 %

2 (16.7)

4 (33.3)

3 (25.0)

3 (25.0)

  1. “In all patients, PH-PVT completely disappeared within 6 months after hepatectomy without invasive surgical thrombectomy or thrombolytic treatment via a catheter positioned in the superior mesenteric artery or portal vein.” This sentence can be shortened to “In all patients, PH-PVT completely disappeared within 6 months after hepatectomy without invasive thrombectomy”

Response: I truly appreciate your valuable feedback. Based on your helpful suggestion, we revised the sentence to align with your recommendations. (line 307-309)

  1. As far as I can see, “postoperaticve complications” were not assessed with their impact on thrombosis? Why? Complication after surgery often lead to extended hospital stay, re-intervtions and increased immobilization, that increase risk of thrombosis.

Response: We appreciate your valuable comments and fully agree with your insights. We conducted a new analysis on the correlation between postoperative complications and PH-PVT. We categorized postoperative complications excluding PH-PVT, into three groups: “none,” “minor” (Clavien-Dindo I or II), and “major” (Clavien-Dindo III or higher). Our findings showed no correlation between postoperative complications and PH-PVT in univariate and stepwise analysis. We have included this factor analysis in Table 4.

We speculate that the development of PVT might be related to specific types of complications. However, because our study had a small sample size, we found it challenging to analyze this in depth. We look forward to future research with a larger sample size.

  1. Extent of steatosis certainly might be relevant. This was not assessed?

Response: Thank you for your valuable feedback. We completely agree and would like to investigate this issue in future multicenter research with a larger sample size. In this study, we found that MASLD was associated with a significantly higher risk of PH-PVT. We suggested the combination of hepatic steatosis and metabolic dysfunction creates a significant procoagulant state that contributes to the development of PH-PVT. As your suggestion, another question is whether the severity of fatty deposition or steatotic hepatitis solely impacts the risk of PVT and its pathophysiological mechanisms. However, our study was a single-center investigation with a small sample size, which limited our ability to analyze the relationship between the extent of steatosis or steatohepatitis and the risk of PVT. Moreover, our institution functions as a clinical hospital rather than a research facility, and we discard resected specimens that are older than five years. As a result, a pathological examination was not feasible for this study. We have noted this issue as a limitation (lines 395-398).

Discussion and Introduction;

  1. I would recommend to stick to definition of e.g. MASLD and not switch between NAFLD and MASLD. It has no merit here to mention the switch/new definition.

Response: We appreciate your important comment. We would like to give our opinion on MASLD research. In 2023, a consensus group from multiple societies redefined the terminology of nonalcoholic fatty liver disease (NAFLD) to metabolic dysfunction-associated steatotic liver disease (MASLD). This change has been supported and endorsed by various organizations worldwide. NAFLD encompasses all forms of steatotic liver disease except those caused by alcohol and therefore, NAFLD is a heterogenous group. On the other hand, the revision to MASLD allows for better capture of the underlying etiologic pathways of the disease. Furthermore, MASLD eliminates potential stigmatization [8,9]

However, as you mentioned, this modification can lead to considerable confusion from various perspectives, and we are likely to face challenges for some time. Therefore, it is crucial to extend previous research on NAFLD to MASLD. Several studies have already provided evidence supporting this transition [10]. While additional research is necessary, we believe the results of this study offer promising suggestions for validating the MASLD classification.

References

  1. Fukunaga S, et al. Impact of non-obese metabolic dysfunction-associated fatty liver disease on risk factors for the recurrence of esophageal squamous cell carcinoma treated with endoscopic submucosal dissection: A multicenter study. Hepatol Res 2024;54:201-212.
  2. Sano T, et al. Metabolic management after sustained virologic response in elderly patients with hepatitis C virus: A multicenter study. Hepatol Res 2024;54:326-335.
  3. Trousseau's syndrome: multiple definitions and multiple mechanisms. Blood 2007; 110:1723-1729.
  4. Strasberg, S.M. Nomenclature of hepatic anatomy and resections: a review of the Brisbane 2000 system. J Hepatobiliary Pancreat Surg 2005;12:351-355.
  5. Kuboki S, et al. Incidence, risk factors, and management options for portal vein thrombosis after hepatectomy: a 14-year, single-center experience. Am J Surg 2015;210:878–885.
  6. Mori A, et al. Risk Factors and Outcome of Portal Vein Thrombosis After Laparoscopic and Open Hepatectomy for Primary Liver Cancer: A Single-Center Experience. World J Surg 2020;44:3093-3099.
  7. Moeini A, et al. Mixed hepatocellular cholangiocarcinoma tumors: cholangiolocellular carcinoma is a distinct molecular entity. J Hepatol 2017;66:952–961.
  8. Rinella ME, et al. A multisociety Delphi consensus statement on new fatty liver disease nomenclature. J Hepatol 2023;79:1542-1556.
  9. Younossi ZM, et al. Global survey of stigma among physicians and patients with nonalcoholic fatty liver disease. J Hepatol 2023;18:S0168–S8278.
  10. Song SJ, Lai JC, Wong GL, Wong VW, Yip TC. Can we use old NAFLD data under the new MASLD definition? J Hepatol 2023;23:S0168–8278.

Reviewer 3 Report

Comments and Suggestions for Authors

There are a few comments.

1. The sample size is small. It would be better to include more cases.

2. A total of 106 cases underwent excision. 

Therefore, it is possible to grade and stage the severity of steatosis (or steatohepatitis). It would be beneficial to include these data.

3. Analyzing portal vein thrombosis according to the underlying tumor types would improve the study.

Comments on the Quality of English Language

Please check Englsih grammar adn spelling.

e.g., necroapoptosis ->necropoptosis

Author Response

To REVIEWER 3

Thank you very much for your letter regarding our manuscript (Cancers-23275601). We appreciate your comments, which have helped us to improve our manuscript. In line with your comments, please find below our point-by-point responses.

  1. The sample size is small. It would be better to include more cases.

Response: As you pointed out, we totally agreed that the sample size of this study was small, especially the MASLD and MetALD (n=20 and n=5). We were not able to increase the sample size immediately. Therefore, we added this issue as a limitation (lines 386-392). However, the following may be possible reasons for the statistically significant differences obtained in this study. MASLD and MetALD groups were relatively homogeneous groups because of their inclusion criteria. By targeting a population with low variability, we might be able to detect effects with a small sample size. In addition, for a large sample size group (No SLD group), we properly handled missing values and outliers to maintain data consistency and improve analysis accuracy. Thus, statistical power might be ensured in this study.

  1. A total of 106 cases underwent excision. Therefore, it is possible to grade and stage the severity of steatosis (or steatohepatitis). It would be beneficial to include these data.

Response: We fully agree that analyzing histological findings related to steatosis or steatohepatitis is valuable. However, our institution functions as a clinical hospital rather than a research facility, and we discard resected specimens that are older than five years. As a result, a pathological examination was not feasible for this study. We include this limitation in our discussion (lines 395-398) and look forward to future research.

  1. Analyzing portal vein thrombosis according to the underlying tumor types would improve the study.

Response: Thank you for your thoughtful suggestion. I agree, and we have included the underlying tumor type—primary liver cancer versus metastatic liver cancer—in our statistical analysis. The results indicated that there was no significant difference between the two, as shown in Table 4. Additionally, factors such as tumor size, number, and the presence of tumor thrombus were also not identified as significant risk factors for PH-PVT. While it would have been interesting to conduct a more detailed analysis of the different tumor types, such as hepatocellular carcinoma versus cholangiocarcinoma, this was not feasible in this study due to the small sample size.

Round 2

Reviewer 3 Report

Comments and Suggestions for Authors

The manuscript was well-revised.

It would be better to describe the abbreviation of MetALD in the Simple Summary section. 

Comments on the Quality of English Language

Please check English grammar and spelling.

  e.g., 50 % -> 50%

       NoSLD -> No SLD

Author Response

Thank you very much for your letter regarding our manuscript (Cancers-23275601). We appreciate your comments, which have helped us to improve our manuscript. In line with your comments, please find below our point-by-point responses.

  1. Simple summary section;

  The term “MetALD” has been revised to “MetALD (MASLD with moderate alcohol consumption).” (lines 17) (Highlighted in yellow)

  1. Quality of English Language

We have reviewed and corrected the grammar and spelling in the text, tables, and figures;

e.g., 50 % -> 50% 

NoSLD -> No SLD

n=6 -> n = 6

2.32 – 37.08 -> 2.32–37.08

         (Highlighted in yellow)

Additionally,

  1. The authors' affiliation and email addresses have been updated (lines 7, 8, 9, 11-12, 13). (Highlighted in yellow)